# DECOMPOSING TEXTURE AND SEMANTICS FOR OUT-OF-DISTRIBUTION DETECTION

## ABSTRACT

Out-of-distribution (OOD) detection has made significant progress in recent years because the distribution mismatch between the training and testing can severely deteriorate the reliability of a machine learning system. Nevertheless, the lack of precise interpretation of the in-distribution limits the application of OOD detection methods to real-world system pipielines. To tackle this issue, we decompose the definition of the in-distribution into texture and semantics, motivated by real-world scenarios. In addition, we design new benchmarks to measure the robustness that OOD detection methods should have. To achieve a good balance between the OOD detection performance and robustness, our method takes a divide-and-conquer approach. That is, the model first tackles each component of the texture and semantics separately, and then combines them later. Such design philosophy is empirically proven by a series of benchmarks including not only ours but also the conventional counterpart.

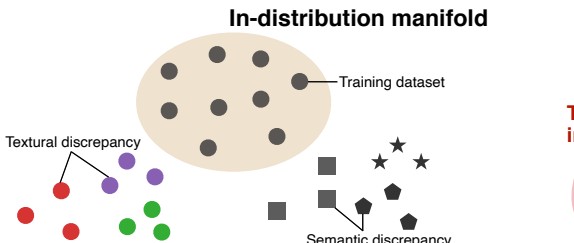 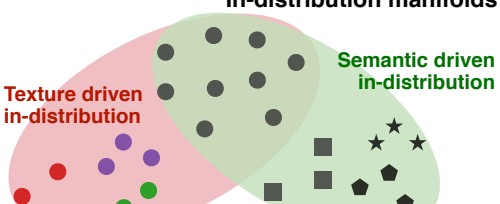

(a) Traditional definition of the in-distribution     (b) Decomposed definition of the in-distribution

Figure 1: **How to define the in-distribution? (a)** Traditional out-of-distribution detection studies have managed the in-distribution in an entangled view. However, this assumption could be naïve considering the complex nature of the real environment. **(b)** We decompose the definition of the in-distribution to the texture and semantic aspects. This provides the flexibility to handle complicated scenarios by determining which definition of the in-distribution is suitable for a given scenario.

## 1 INTRODUCTION

The out-of-distribution (OOD) detection is the task that recognizes whether the given data comes from the distribution of training samples (also known as *in-distribution*) or not. Any machine learning-based system could receive input samples that have a completely disparate distribution from the training environments (*e.g.,* dataset). Since the distribution shift can severely degrade the model performance (Amodei et al., 2016), it is a potential threat for a reliable real-world AI system.

However, an ambiguous definition of the "in-distribution" limits the feasibility of the OOD detection method in real-world applications, considering the various OOD scenarios. For example, subtle corruption is a clear signal of the OOD in the machine vision field while a change in semantic information might not be. On the other hand, an autonomous driving system may assume the in-distribution from the semantic-oriented perspective; *e.g.,* an unseen traffic sign is the OOD. Interestingly, the interpretations of the in-distribution described in the above scenarios are not correlated; rather, they are contradicted. Unfortunately, most of the conventional OOD detection methods (Zhang et al.,

2021; Tack et al., 2020; Ren et al., 2019) assume the in-distribution as a single-mode; thus are not able to handle other aspects of the OOD (Figure 1a).

To tackle this issue, we revisit the definition of the in-distribution by decomposing it into two different factors: *texture* and *semantics* (Figure 1b). For the texture OOD case, we define the OOD as the textural difference between the in- and out-of-distribution datasets. On the contrary, the semantic OOD focuses on the class labels that do not exist in the in-distribution environment. Note that the two aspects have a trade-off relationship, thus detecting both OOD problems with a single model is challenging with the (conventional) entangled OOD point of view.

Similar to ours, Geirhos et al. (2018) investigated the texture-shape cue conflict in the network, and a series of following studies (Hermann et al., 2019; Li et al., 2020; Ahmed & Courville, 2020) explored the way to find a balance between these perspectives. However, aforementioned works utilize texture-shape to analyze the bias inherited in deep networks. In this work, instead, we focus on analyzing the texture and semantic nature underlying the in-distribution to build a more practically applicable OOD detection method.

However, to the best of our knowledge, none of the studies on the OOD detection benchmark have thoroughly investigated the definition of the in-distribution. This can be problematic when the OOD detection method judges the image corrupted by minor distortion as OOD, even when the environment is tolerant to the small changes in texture. Because of such a complicated scenario, it is important to evaluate the OOD detection method in a comprehensive way that goes beyond the simple benchmarks. In this work, we propose a new approach to measuring the performance of the method according to the decomposed definition of the in-distribution. One notable observation in our benchmark is that most of the previous OOD detection methods are highly biased to the texture information and ignore the semantic clues in many cases.

To mitigate this issue, our method tackles the texture and semantic information separately and aggregates these at the final module (Figure 2). To effectively extract the texture information, we use the 2D Fourier transform motivated by the recent frequency domain-driven deep method (Xu et al., 2020). For the semantic feature, we design an extraction module upon the Deep-SVDD (Ruff et al., 2018) with our novel angular distance-based initialization strategy. We then combine two features using the normalizing flow-based method (Dinh et al., 2016), followed by our factor control mechanism. This control module provides the flexibility to handle various OOD scenarios by choosing which decomposed feature is more important in the given surrounding OOD circumstance.

The main contributions of this work are as follows:

- We decompose the "unclear" definition of the in-distribution into *texture & semantics*. To the best of our knowledge, this is the first attempt to clarify the OOD itself in this field.

- Motivated by real-world problems, we create new OOD detection benchmark scenarios to evaluate the models based on the decomposed in-distribution factors.

- We propose a novel OOD detection method that is effective on both *texture & semantics* as well as the conventional benchmark setups. In addition, our method does not require any auxiliary datasets or class labels unlike the previous models.

## 2 RELATED WORK

In this section, we briefly overview the notable studies on the OOD detection field. We categorize the deep learning-based OOD detection methods into three groups based on the characteristics of the information that they used.

**Class labels of the in-distribution.** Early studies on deep OOD methods rely on class supervision. ODIN, Generailized ODIN (Liang et al., 2017; Hsu et al., 2020) use the uncertainty measure derived by the Softmax output. It determines the given sample as the OOD when the output probability of all classes is less than the predefined threshold. Sastry & Oore (2020); Lee et al. (2018) utilize the extracted feature map (*e.g.,* gram matrix) from the pre-trained networks to calculate the OOD score.

**Auxiliary distribution.** Outlier exposure (OE) (Hendrycks et al., 2018) exploits additional datasets that are disjointed from the test dataset to guide the network to better representations for OOD

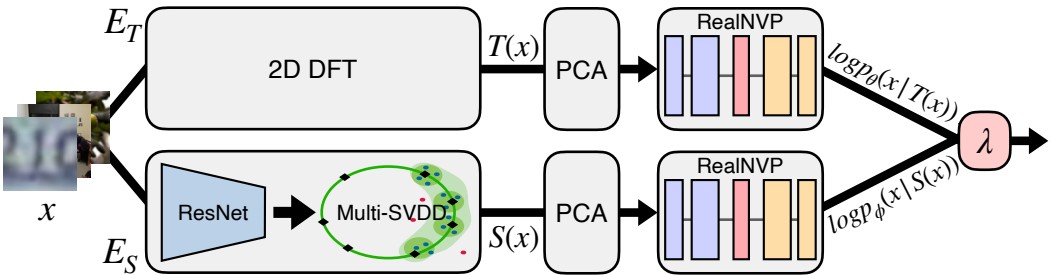

Figure 2: **Model overview.** Our framework first extracts the texture and semantic information with the corresponding modules, and then combines them via the normalizing flow-based method. **(a)** Texture feature $T(x)$ is distilled by the Fourier spectrum-based component. **(b)** We use a multi-SVDD method with a novel angular initialization to extract the semantic information $S(x)$. **(c)** Output features are merged by the explicit probability inference method, RealNVP. Here, we introduce the user control parameter $\lambda$ to determine which feature is more suitable for a given OOD scenario.

detection. Papadopoulos et al. (2021) further improves the performance of OE by regularizing the network with the total variation distance of the Softmax output.

**Data augmentation.** Recently, contrastive learning-based methods have shown remarkable success on the tasks related to visual representation (He et al., 2020; Chen et al., 2020). Motivated by this, several studies employ data augmentation methods such as image transformation or additional noise on the OOD detection task (Hendrycks et al., 2019; Tack et al., 2020).

Unlike the prior studies that exploit additional information other than the in-distribution, we only utilize the given (in-distribution) training dataset. In addition, we separate and clarify the assumption of the OOD as texture and semantics to improve the practicability in the real world.

## 3 METHOD

In this section, we present an overview of our proposed method (Section 3.1), and the feature extraction modules of the model (Section 3.3 and 3.2). Finally, we introduce the normalizing flow-based conditional probabilistic modeling component (Section 3.4). Conventional OOD detection has an assumption that in-distribution data are sampled from the distribution of the training dataset, $x \sim p_{data}$. We decompose the image with two factors and calculate the anomaly score based on each factor's likelihood. The texture information T(x) extracted rigorous Fourier analysis process from input x. The semantic information S(x) extracts the content label features such as shape. Our framework calculates the likelihood of these two factors and then combines these likelihoods. Since we use the Normalizing flow model that trains the exact likelihood, the extracted information is adjusted using a lambda that the user can control.

### 3.1 MODEL OVERVIEW

We aim to train our method with the decomposed in-distribution likelihood, $p(T(x)|x)$ and $p(S(x)|x)$ (Figure 2). With the given input image $x$, we extract the features for each variable with different approaches. In detail, we distil the information from the texture and semantic information with $T(x)$ and $S(x)$, respectively.

The extracted features are combined by the controllable normalizing flow method. Since our normalizing flow-based model explicitly calculates the negative log-likelihood, we model each extracted information as $\log p_\theta(T(x)|x)$ and $\log p_\phi(S(x)|x)$, where $\theta$ and $\phi$ are trainable parameters of the networks. In addition, we introduce the control parameter $\lambda \in [0, 1]$ to model the final probability as $\lambda \cdot \log p(x|T(x)) + (1 - \lambda) \cdot \log p(x|S(x))$. With this control mechanism, a user can determine the appropriate model "mode" by referring to the prior knowledge. For example, in the case where the texture information overwhelms the semantic one for detecting OOD, we can overweight $\lambda$ for better performance. By default, we use the $\lambda$ value of 0.5 (no prior knowledge).

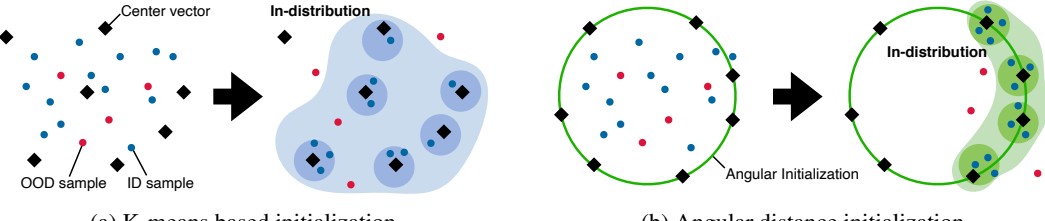

(a) K-means based initialization          (b) Angular distance initialization

Figure 3: **Comparison on the initialization strategy in multi-SVDD. (a)** The model with K-means initialization is effective for the anomaly detection but not for the OOD detection scenario. This is because the definition of the anomaly is the samples that do not belong to the cluster regions (dark shade), while the definition of the OOD is the samples that does not lie in the in-distribution manifold (light shade). **(b)** Our proposed angular distance-based initialization guides the initial center vectors to be positioned following a (virtual) circular line. As a result, it prevents the OOD samples from not belonging to the in-distribution by creating tight cluster layouts without a hole.

## 3.2 EXTRACTING THE SEMANTIC INFORMATION

**Multi-SVDD.** Beyond the one-class anomaly detection that considers the normal data as a single class (*e.g.,* DeepSVDD (Ruff et al., 2018)), recent studies have viewed the normal data as the union of the multiple hidden semantic information (Ghafoori & Leckie, 2020; Park et al., 2021). Inspired by this idea, we use the multi-SVDD method to extract the semantic information in an unsupervised manner for the OOD detection task.

Multi-SVDD embeds the samples to the multiple center vectors as close as possible. Suppose the set of center vectors $\mathbf{C} = \{c_1, ..., c_K\}$ is initialized via K-means and the radius of each center is $r = [r_1, ..., r_K]$. In multi-SVDD, the objective function is defined as follows.

$$\min_{\mathcal{W},\mathbf{r}} \quad \sum_{k=1}^{K} r_k^2 + \frac{1}{\nu n} \sum_{i=1}^{n} \max\left\{0, \|\phi(x_i; \mathcal{W}) - c_j\|^2 - r_j^2\right\} + \frac{\eta}{2} \sum \|\mathcal{W}\|^2. \tag{1}$$

Here, $\phi(x_i; \mathcal{W})$ is the deep network with a set of weight parameters $\mathcal{W}$ and $c_j$ is assigned to $\phi(x_i; \mathcal{W})$. As the set $r$ is decreased, the samples are condensed into the center vectors. By using the distance between the center vectors and the samples, we get an anomaly score.

**Angular distance initialization.** The SVDD method is originally introduced for the anomaly detection task. Because of the disparity between the OOD and the anomaly detection scenarios, direct application of the SVDD-based model to OOD detection causes unexpected performance degradation. In anomaly detection, even though the abnormal samples lie in the in-distribution manifold, it is possible to detect them as abnormal unless they are close to the center vectors $c_j$. For example, as shown in Figure 3a, the OOD samples (red) that are located inside of the in-distribution manifold (light blue shade) can be detected as abnormal since they are outside of the tight cluster boundary (dark blue shade). Because of such characteristics, a mixture of Gaussian probability density is a reasonable density space for the anomaly detection model.

Unlike the anomaly detection task, the definition of the OOD detection task is to find the samples that are not the "in-distribution". In Figure 3a, all the OOD samples placed in the in-distribution manifold (light dark shade) are recognized as the in-distribution. To tackle this issue, we propose an angular distance-based center vector initialization strategy:

$$c_k = \gamma \frac{\mathbf{v}}{\|\mathbf{v}\|}, \quad \mathbf{v} \in \mathbb{R}^h \sim \mathcal{N}(0, 1) \tag{2}$$

where $h$ is the dimension of the embedding space and $\gamma$ is the hyper-parameter for the radius of the sphere. After $\phi(x_i; \mathcal{W})$ is trained based on angular initialization, semantic features are extracted through this model; $S(x_i) = \phi(x_i; \mathcal{W})$. By setting the $\gamma$ value as large enough, we ensure that all sample data are within a radius of the sphere as illustrated in Figure 3b. While Equation 1 drives the training samples to be embedded around the center vectors on the sphere, the OOD samples remain near the origin. This embedding space may be weak to recognize the semantic label of a given sample, but it is sufficient to identify whether the sample is OOD or not.

### 3.3 EXTRACTING THE TEXTURE INFORMATION

To effectively extract the texture property of the in-distribution, we interpret the image in the frequency space. With a given input image $x \in \mathbb{R}^{3 \times h \times w}$, we first convert it into the frequency domain using Discrete Fourier Transform (DFT) $\mathcal{F}$ as shown below.

$$\mathcal{F}(f_x, f_y) = \frac{1}{hw} \sum_{p=0}^{h-1} \sum_{q=0}^{w-1} I(p, q) \cdot e^{-i2\pi(f_x p/h + f_y q/w)}, \tag{3}$$

Here, $I(p, q)$ denotes the pixel value of the image at the $(p, q)$-coordinate and $\mathcal{F}(f_x, f_y)$ is the output of the DFT at the Cartesian coordinate $(f_x, f_y)$ in the frequency space. In order to construct a scale and rotation invariant frequency information in 2D image, we modify the coordinate system from Cartesian $(f_x, f_y)$ to polar $(f_r, \theta)$, following Dzanic et al. (2019).

$$\mathcal{F}(f_r, \theta) = \mathcal{F}(f_x, f_y) \quad : \quad f_r = \sqrt{\frac{f_x^2 + f_y^2}{\frac{1}{4}(m^2 + n^2)}}, \quad \theta = \operatorname{atan2}(f_y, f_x). \tag{4}$$

Since directly computing the polar coordinate is computationally expensive and tricky, we iteratively calculate the rotation invariant frequency feature. To do that, we only utilize the first channel of the image as $x \in \mathbb{R}^{1 \times h \times w}$ and assume that the image is square (*i.e.*, $h = w$). Let $T(x) \in \mathbb{R}^{2/w}$ be the texture feature vector of the image $x$. Then, the $i$-th element of the feature $T_i(x)$ is calculated as:

$$T_i(x) = R_i - R_{i-1} \quad : \quad R_w = \sum_{f_x=-w}^{w} \sum_{f_y=-w}^{w} \mathcal{F}(f_x, f_y), \quad R_0 = \mathcal{F}(0, 0) \tag{5}$$

**Discussion.** We compare the power spectrum density (PSD) of CIFAR-10 (Krizhevsky et al., 2009), distorted CIFAR-10 and CIFAR-100 (Krizhevsky et al., 2010) datasets (Figure 4). The PSD discrepancy between the corrupted CIFAR-10 and the vanilla one indicates that the image feature acquired from the frequency domain is adequate to represent the texture cue. In contrast, CIFAR-10 and CIFAR-100 are not distinguishable in the frequency domain since they have very similar image texture due to the small resolution. These observations support our assumption that the OOD detection model should decouple the texture and semantic information to properly handle the uncorrelated cues.

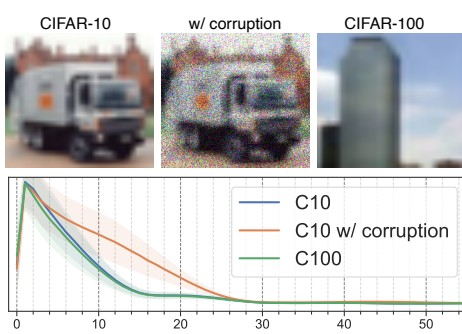

Figure 4: **Power spectrum density analysis.**

### 3.4 FEATURE COMPOSITION VIA NORMALIZING FLOW

Since we design our framework to directly sample the probability, any ad-hoc scoring functions are not required. Instead, we use a normalizing flow-based method (Dinh et al., 2014; Rezende & Mohamed, 2015; Dinh et al., 2016) that uses the probability of given samples as a loss function. In the following, we will describe how to get the probability of samples from the prior probability (Normal distribution) using normalizing flow.

Given sample $x$, a normal prior probability distribution $p_Z$ on a latent variable $z \in Z$, and a bijection $f : X \to Z$ (with $g = f^{-1}$), the change of variables defines a model distribution on $X$ by

$$p_X(x) = p_Z\big(f(x)\big) \left| \det\left( \frac{\partial f(x)}{\partial x^T} \right) \right|, \tag{6}$$

where $\frac{\partial f(x)}{\partial x^T}$ is the Jacobian of $f$ at $x$. The bijection function $f$ can be decomposed as $f = f_1 \circ \cdots \circ f_k$.

We use a flow-based RealNVP with coupling layers (Dinh et al., 2016). To provide the input to RealNVP, we apply PCA for the dimension reduction of each extracted features. Finally, given the

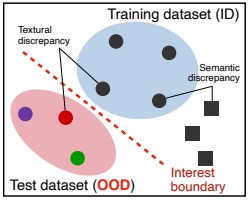 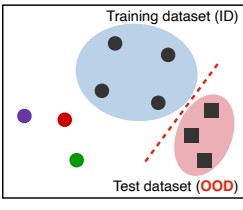 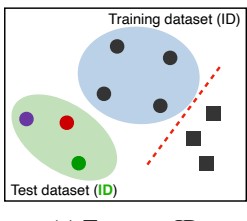 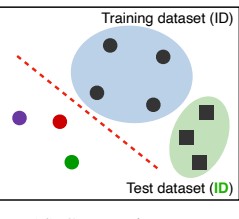

| (a) Texture - OOD | (b) Semantics - OOD | (c) Texture - ID | (d) Semantics - ID |

Figure 5: **Comparison of the benchmark scenarios. (a, b)** Out-of-distribution detection scenario. The training dataset is in-distribution (ID), while test datasets are out-of-distribution (OOD). Note that texture and semantic OODs are entangled in the conventional benchmark. **(c, d)** Evaluation on the robustness. Unlike (a, b), it provides the ID samples at test time. An oracle in this scenario produces AUC as 50.0 (cannot distinguish as OOD at all) since the given samples are from ID.

in-distribution training dataset $\mathcal{D}$, the objective of the RealNVP model with the trainable parameters $\theta$ and $\phi$ is to minimize the following equations.

$$\mathcal{L}_{texture}(\mathcal{D}) = \frac{1}{N}\sum_{i=1}^{N} -\log p_\theta(T(x_i)|x_i), \quad \mathcal{L}_{semantics}(\mathcal{D}) = \frac{1}{N}\sum_{i=1}^{N} -\log p_\phi(S(x_i)|x_i) \quad (7)$$

By decomposing to the texture and semantic components based on probabilistic modeling, one nice side-effect is that we can adjust the contribution of these components by referring to the given OOD environment. Since we designed the feature components to be separate, we combine them with a simple linear interpolation as follows.

$$\lambda \cdot \log p_\phi(S(x)|x) + (1-\lambda) \cdot \log p_\theta(T(x)|x), \quad (8)$$

where $\lambda \in [0, 1]$ is the control parameter (default is 0.5).

## 4 EXPERIMENTS

In this section, we demonstrate the effectiveness of our proposed method on various benchmark setups. In Section 4.1, we report the performance on the conventional OOD detection task and discuss the limitation of the previous OOD detection studies. We use the area under the curve (AUC) for the receiver operating characteristic (ROC) curve to evaluate the OOD detection performance. Then, we will discuss on how to evaluate the robustness that the OOD detection method should have and show the robustness of the previous and ours models (Section 4.2).

### 4.1 CONVENTIONAL OUT-OF-DISTRIBUTION DETECTION

**Setups.** We evaluate out-of-distribution detection methods on the widely used OOD detection benchmark. Here, we use SVHN (Netzer et al., 2011), CIFAR-10 (Krizhevsky et al., 2009) and CIFAR-100 (Krizhevsky et al., 2010) as the in-distribution dataset. To simulate the OOD samples, LSUN (Yu et al., 2015) and Tiny-ImageNet (Torralba et al., 2008) datasets are additionally used.

**Baselines.** We compare our approach to the methods belonging to three different OOD detection groups. **1)** Methods that use class labels of the training samples. Feature based methods such as Maha (Lee et al., 2018) and Gram (Sastry & Oore, 2020) fall into this category. **2)** Methods that utilize additional distribution (dataset) such as OE (Hendrycks et al., 2018) and OEC (Papadopoulos et al., 2021). **3)** Self-supervised based methods. Rotation-based (Rot) (Hendrycks et al., 2019), SSL (Mohseni et al., 2020), and CSI (Tack et al., 2020) are in this group.

**Results.** As shown in Table 1, our proposed method surpasses the competitors on the conventional OOD detection task without using any extra information such as class labels, other datasets, or image transformation techniques that the other methods required. In detail, ours with $\lambda = 1.0$ (semantic mode) achieves the best performance in all the cases with the exception of two scenarios.

**Discussion.** As we have discussed, it is often risky to assume that the input samples of the real-world system have a single and entangled characteristic. We argue that it is more natural to consider

| ID → OOD | | Using Labels | | Other Dist. | | Self-supervised | | | Ours λ = | | |
|---|---|---|---|---|---|---|---|---|---|---|---|
| | | Maha | Gram | OE | OEC | Rot | SSL | CSI | 0.0 | 0.5 | 1.0 |
| **SVHN** | C10 | 99.3 | 97.3 | 99.3 | 99.8 | - | 99.8 | - | 93.9 | 99.9 | **100.** |
| | C100 | - | - | 99.0 | **99.9** | - | 99.8 | - | 91.1 | 99.8 | **99.9** |
| | TinyImgNet | 99.3 | 97.3 | - | - | - | - | - | 99.5 | **100.** | **100.** |
| | LSUN* | 99.9 | 99.8 | 99.9 | 99.9 | - | 99.9 | - | 99.9 | **100.** | **100.** |
| **C10** | SVHN | 99.1 | 99.5 | 98.2 | 99.2 | 97.8 | 99.2 | 99.8 | 86.1 | **99.9** | **99.9** |
| | C100 | 88.2 | 79.0 | 92.9 | **93.8** | 82.3 | **93.8** | 89.2 | 55.7 | 93.6 | 93.5 |
| | TinyImgNet | 99.5 | 99.7 | - | - | - | - | - | 65.4 | **99.9** | **99.9** |
| | LSUN* | 99.3 | **99.9** | 96.4 | 98.9 | 92.8 | 98.9 | 97.5 | **99.9** | 98.8 | 81.9 |
| **C100** | SVHN | 98.4 | 97.3 | 82.8 | 95.8 | - | 95.8 | - | 80.0 | 99.9 | **100.** |
| | C10 | 77.5 | 67.9 | 77.5 | 77.7 | - | 77.7 | - | 49.4 | 83.1 | **84.2** |
| | TinyImgNet | 97.4 | 99.0 | - | - | - | - | - | 91.7 | **100.** | **100.** |
| | LSUN* | 98.2 | 99.3 | 79.5 | 88.8 | - | 88.8 | - | 99.9 | **100.** | **100.** |

Table 1: **Conventional OOD detection benchmark.** We evaluate the detection performance by AUC (in- vs. out-distribution detection based on confidence/score) in percent (**higher is better**). C10 and C100 stand for the CIFAR-10 and CIFAR-100. * indicates the high-resolution dataset.

that the data samples can have multiple attributes such as texture and semantic information (Figure 5a and 5b). For example, when the given environment requires to detect the textural discrepancy as OOD, then the detection method should concentrate on the textural side alone, not the semantic counterpart, and vice versa for the semantics driven OOD case.

Unfortunately, conventional OOD detection benchmarks are not developed to measure the disentangled view of the in-distribution. Here, we dissect the traditional benchmark to quantify the effect of each property we decompose. We first categorized the datasets into the high- and low-resolution groups by the image resolution: C10, C100, SVHN, and TinyImageNet belong to the low-resolution (LR) group while LSUN is high-resolution (HR). Table 1 shows that previous studies achieve high performance on the LR → HR scenarios and label-based methods produce outstanding results. However, in Section 4.2.1, we will show that the superior performance of these methods is because they abuse the texture information (*e.g.,* detect as OOD by highly referring to the image resolution).

On the other hand, we argue that the semantic property is the key component to identity the OOD of the LR → LR scenarios. For example, C10 ↔ C100 solely requires semantic information to detect OOD since these datasets use very similar images (in terms of the texture and image resolution). This is the reason why ours with $\lambda = 0.0$ (texture mode) is not able to detect OOD at all (55.7 and 49.4 AUC). Note that other competitors also show the inferior performance, especially C10 ↔ C100 case, which demonstrate that these methods have weakness in handling semantic information.

## 4.2 EVALUATING THE ROBUSTNESS OF THE OUT-OF-DISTRIBUTION DETECTION METHOD

In this section, we evaluate the robustness of the OOD detection method. The following experiments are motivated by the real-world demands that the OOD detection framework focused on the semantic discrepancy should be tolerant to the minor changes in the image space (such as the texture shown in Figure 5c), and vice versa (Figure 5d). However, how can we evaluate the robustness of the OOD detection method in texture and semantic perspectives?

To quantify the robustness of the OOD method, we introduce the AUC score-based measurement. The evaluation protocol is as follow: **1)** We first train the OOD detection method with the in-distribution training dataset. **2)** We provide the in-distribution samples as test dataset. Here, contrary to the conventional benchmark, AUC $= 50\%$ is the best performance. This is because 50% AUC indicates that the detection method cannot distinguish the test samples as OOD (determines as in-distribution). In Section 4.2.1, we evaluate the robustness on the texture discrepancy by providing the textually different (but marginally) in-distribution dataset at test time (Figure 5c). Then, in Section 4.2.2, we benchmark the robustness on the semantic discrepancy (Figure 5d).

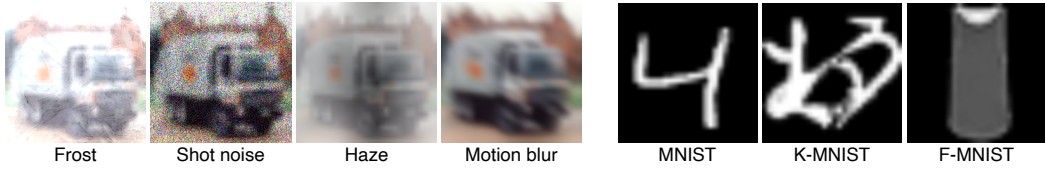

| (a) CIFAR-10 w/ corruptions | (b) MNIST variants |

Figure 6: **Example of the datasets used in our proposed benchmark. (a)** Corrupted CIFAR-10 dataset to evaluate the robustness on the texture discrepant in-distribution case (Section 4.2.1). **(b)** MNIST variants datasets for the semantic discrepancy scenario (Section 4.2.2).

| Level | 1 | 2 | 3 | 4 | 5 |
|---|---|---|---|---|---|
| ODIN | 67.4 | 71.9 | 78.6 | 79.4 | 80.5 |
| Maha | 79.3 | 89.8 | 95.9 | 96.2 | 98.2 |
| Gram | 99.7 | 99.9 | 99.9 | 99.9 | 99.9 |
| CSI | 65.1 | 73.3 | 81.1 | 81.8 | 86.9 |
| 0.0 | 58.1 | 68.8 | 79.7 | 80.3 | 86.5 |
| $\lambda = 0.5$ | 58.1 | 62.2 | 72.7 | 77.9 | 82.9 |
| 1.0 | **56.8** | **54.9** | **57.8** | **63.9** | **70.8** |

(a) Frost

| Level | 1 | 2 | 3 | 4 | 5 |
|---|---|---|---|---|---|
| ODIN | 76.0 | 74.5 | 75.3 | 71.0 | 68.0 |
| Maha | 99.0 | 99.1 | 99.8 | 99.8 | 99.9 |
| Gram | 99.0 | 99.9 | 99.9 | 99.9 | 99.9 |
| CSI | 54.6 | 59.8 | 66.1 | 74.1 | 89.2 |
| 0.0 | 50.1 | 59.9 | 68.1 | 89.0 | 99.9 |
| $\lambda = 0.5$ | 50.1 | 56.6 | 67.4 | 74.9 | 72.2 |
| 1.0 | **49.9** | **57.7** | **59.9** | **64.4** | **69.9** |

(b) Haze

| Level | 1 | 2 | 3 | 4 | 5 |
|---|---|---|---|---|---|
| ODIN | 63.1 | 66.0 | 72.4 | 79.9 | 82.0 |
| Maha | 84.9 | 88.8 | 95.9 | 96.9 | 97.9 |
| Gram | 99.8 | 99.9 | 99.9 | 99.9 | 99.9 |
| CSI | 67.7 | 77.5 | 85.1 | 85.1 | 90.1 |
| 0.0 | 67.3 | 81.3 | 88.6 | 88.9 | 93.1 |
| $\lambda = 0.5$ | 64.9 | 66.0 | 83.3 | 88.6 | 93.0 |
| 1.0 | **51.1** | **52.1** | **61.4** | **61.6** | **64.7** |

(c) Motion blur

| Level | 1 | 2 | 3 | 4 | 5 |
|---|---|---|---|---|---|
| ODIN | 69.9 | 75.5 | 76.4 | 81.2 | 87.2 |
| Maha | 95.8 | 98.6 | 99.7 | 99.8 | 99.9 |
| Gram | 99.5 | 99.8 | 99.9 | 99.9 | 99.9 |
| CSI | 77.5 | 85.5 | 94.5 | 96.2 | 98.1 |
| 0.0 | 68.1 | 70.2 | 79.3 | 89.9 | 92.3 |
| $\lambda = 0.5$ | 59.7 | 63.3 | 68.4 | 79.3 | 85.7 |
| 1.0 | **56.8** | **59.2** | **63.6** | **70.0** | **78.3** |

(d) Shot noise

Table 2: **Robustness on the texture discrepancy produced by the mild corruptions.** We use CIFAR-10 as the in-distribution dataset, and corrupt this using mild distortions. In this benchmark, an OOD detection method should not determine the corrupted images as OOD (*i.e.,* **50.0% AUC is the best score**). This scenario is motivated by the real-world applications that context information is the key factor in OOD but the mild corruptions are acceptable.

### 4.2.1 ROBUSTNESS ON TEXTURE DISCREPANCY OF THE IN-DISTRIBUTION

**Setups.** Here, we experiment on two benchmark scenarios: **1)** distortion and **2)** image resolution. Both cases assume that the definition of the OOD is in the disparity of semantic information. That is, a mild change presented in the image space should not be considered as a signal of the OOD.

For the distortion scenario, we corrupt the CIFAR-10 dataset with frost, shot noise, haze, and motion blur distortions with diverse (but mild) corruption levels (Figure 6a). To make the resolution change case, we first perform center-crop and resize it back to the original resolution ($32\times32$). We carefully adjust the cropping operation not to harm the semantic information.

**Result.** Table 2 shows the AUC score of the OOD detection. Since this experiment focuses on detecting the test samples as the in-distribution, 50.0% AUC is the best score. We observe that the deep feature-based methods (Maha and Gram) do not have robustness against the texture discrepancy at all. They are not appropriate for real-world applications that do not treat a mild corruption as OOD. On the contrary, our method with $\lambda = 1.0$ (semantics mode) achieves the best performance in all the scenarios because this concentrates on the semantic information alone.

| Method | Resolution $32^2 \to$ | | | |
|---|---|---|---|---|
| | $36^2$ ($\downarrow$) | $40^2$ ($\downarrow$) | $44^2$ ($\downarrow$) | $48^2$ ($\downarrow$) |
| ODIN | 55.7 | 62.7 | 67.5 | 74.2 |
| Gram | 66.5 | 74.8 | 82.4 | 92.9 |
| CSI | 81.5 | 86.1 | 89.9 | 93.5 |
| 0.0 | 59.9 | 67.7 | 75.8 | 82.8 |
| $\lambda = 0.5$ | 50.2 | 55.8 | 71.2 | 72.8 |
| 1.0 | **50.1** | **55.0** | **60.9** | **66.0** |

Table 3: **Robustness on the texture discrepancy.** We use $32\times32$ CIFAR-10 as the ID and vary the resolution to make test datasets. Note that **50.0 is the best** score since the test datasets are from the in-distribution but with different image resolutions.

| Method | MNIST $\to$ | |
|---|---|---|
| | K-MNIST ($\downarrow$) | F-MNIST ($\uparrow$) |
| Maha | 91.6 | 96.8 |
| OE | 97.6 | 99.8 |
| SSL | 99.9 | **100.** |
| 0.0 | **50.6** | 91.8 |
| $\lambda = 0.5$ | 100. | **100.** |
| 1.0 | 100. | **100.** |

Table 4: **Robustness on the semantic discrepancy.** In the K-MNIST scenario, **50.0 is the best** since no textural difference exists between the ID and OOD, in contrast to the F-MNIST (**higher is better**).

In the resolution change scenario (Table 3), our method with $\lambda = 1.0$ (semantics mode) outperforms the others in all the settings. All the methods except ours with semantics mode are extremely sensitive to the image resolution change, although no other information is modified.

### 4.2.2 ROBUSTNESS ON SEMANTIC DISCREPANCY OF THE IN-DISTRIBUTION

**Setups.** To evaluate the robustness that may arise from the semantic discrepancy, we use MNIST, K-MNIST, and F-MNIST datasets. K-MNIST comprises of Japanese characters while F-MNIST is a collection of fashion objects. In this experiment, we set both MNIST and K-MNIST as the in-distribution datasets since they are similar in terms of the texture aspect (only the character labels are different, as shown in Figure 6b). On the other hand, MNIST $\leftrightarrow$ F-MNIST is the conventional OOD scenario since they have both disparate semantics and textures.

**Result.** In Table 4, we compare our method with the OOD methods that use class labels or with a self-supervision based approach. For MNIST $\to$ K-MNIST scenario, our method with $\lambda = 0.0$ successfully determines that K-MNIST is the in-distribution (*e.g.,* AUC is 50.6). In contrast, our model with other $\lambda$ values (0.5 and 1.0) cannot, since these mostly depend on the semantic information to detect the OOD. This behavior also appears in the other OOD detection methods because they use entangled information. Not surprisingly, all the methods identify that given test samples are OOD in the MNIST $\to$ F-MNIST case, where the texture information is the crucial component. A series of experiments prove that the texture and semantic cues should be viewed as disentangled factors.

## 5 CONCLUSION

In this work, we introduce a novel viewpoint of the in-distribution for the practically applicable OOD detection in the real world. To do that, we separate the definition of the "single-mode" in-distribution to the "texture" or "semantics" factors by following the requirements of the real-world applications. To effectively handle both aspects, we take a divide-and-conquer strategy that extracts the features using the appropriate method in each factor, then combines these with the normalizing flow-based model. By doing so, our method outperforms previous models on not only our newly proposed benchmark scenarios but also the conventional OOD detection cases.

We hope that our work provides useful guidance for future OOD detection work. In the future, we aim to investigate more diverse factors beyond the textures and the semantics that complicated datasets can have, such as multi-object scenes.

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
