# OpenReview forum: "Decomposing Texture and Semantics for Out-of-distribution Detection"
_ICLR.cc/2022/Conference — ICLR 2022 Submitted_

### Official Review · Reviewer_nEb7 · 2021-11-02

**Correctness:** 4
**Technical Novelty And Significance:** 3
**Empirical Novelty And Significance:** 3
**Recommendation:** 6
**Confidence:** 3

**Main Review:**

## Strength
### 1. The paper is well organized and written, such that it is easy to follow.
### 2. The proposed texture and semantic decomposition idea is technically sound and correct. The motivation of the idea is clearly explained. The experiment is extensive and sufficient to justify the proposed idea.

##Weakness
### 1. The paper assumes that the texture and semantic information are independent. In my understanding, this assumption is a little strong. How to make sure the assumption is valid or when the assumption is valid. This may need some justifications.


**Summary Of The Paper:**

The paper propose to decompose the out of distribution sample detection task into two-fold: texture and semantic. Extensive discussions and experiments are presented to demonstrate the efficacy of the proposed decomposition idea.

**Summary Of The Review:**

Overall the paper is good and ready for publication.

---

> ### Author Response · Authors · 2021-11-17
> **Response to reviewer nEb7**
>
> __R4-Q1) Independence on the texture and semantics is too strong.__
>
> R4-A1) We appreciate your comment. This feedback was also raised by R1-Q1. We relax the previous assumption to focus on the “decomposing” perspective only. We would like to emphasize that our framework and implementation are not changed even if we relax the assumption. Please see R1-A1 for more details.

---

### Official Review · Reviewer_7WrH · 2021-11-02

**Correctness:** 4
**Technical Novelty And Significance:** 2
**Empirical Novelty And Significance:** 3
**Recommendation:** 6
**Confidence:** 3

**Main Review:**

Strengths:
- The separation of texture and semantics is reasonable
- SoA performance
- Easy-to-understand writing and illustration

Questions and weaknesses:
- Is there any case in which texture is part of semantics?
- How do we know the optimization of (2) will extract semantic information but not the texture information?
- It looks like the method with only semantics (no texture enforced, lambda=1.0) works the best in most cases. Is it still important to enforce texture information extraction?
- Lack of comparison to latest methods (e.g. [A], [B], [C])
[A] Zisselman et al., Deep Residual Flow for Out of Distribution Detection, CVPR20
[B] Lin et al., MOOD: Multi-level Out-of-distribution Detection, CVPR21
[C] Zaeemzadeh et al., Out-of-Distribution Detection Using Union of 1-Dimensional Subspaces, CVPR21


**Summary Of The Paper:**

The paper proposes an OOD setting emphasizing on texture and semantics. The authors propose an OOD detection method which disentangles texture and semantics. The method achieves SoA performances.

**Summary Of The Review:**

On the good side, the proposal of OOD problem with texture and semantics is reasonable and the proposed method achieves SoA performance. However it still lacks comparison with some latest methods.

---

> ### Author Response · Authors · 2021-11-17
> **Response to reviewer 7WrH**
>
> __R3-Q1) Is there any case in which texture is part of semantics? And how do we ensure that Eq 2 only extracts semantics?__
>
> R3-A1)  We appreciate your valuable comment. The texture could be part of semantics. The horse and zebra case (as commented by R1-Q2) is one possible case. We also agree that the extracted features from the semantics module partially contain texture information. This may arise because of 1) the coupling nature of the dataset (as the horse-zebra case) or 2) the limited capability of the extracted module.
>
> For the first case, we admit that we cannot guarantee the perfection of disentanglement of the semantics, and we believe that most of the studies on the dataset bias or feature disentanglement have similar difficulties. Thus, in this study, we focused on decomposing the semantics and texture to give controllability and flexibility to a user by providing the decoupled two steering wheels (i.e. factors), not to handle the case where texture is part of semantics.
>
> However, other than the first case, we improve the previous semantics module (multi-SVDD) to better extract the semantic information by a novel angular initialization strategy. In Figure 2 in Appendix, the figure visualizes the embedding results of the multi-SVDD with our novel initialization methods. The semantic module with our initialization only separates the semantically different datasets (C10 vs. SVHN), while not so sensitive for the texturally-disparate case (C10 vs. corrupted C10). This result implies that although semantics and textures cannot be decoupled completely (because of the nature of some datasets), our framework is effective in many OOD cases.
>
> __R3-Q2) Importance of the texture information.__
>
> R3-A2) As an example of Table 1, since the (previous) definition of the OOD is highly related to the semantics (e.g. different dataset to the training one is the OOD), it is not surprising that semantics mode outperforms in this case. However, texture corruption or some real-world scenarios (as described in R1-A2) require the concentration of the texture information.
>
> __R3-Q3) Lack of comparison to latest methods.__
>
> R3-A3) Thank you for letting us know about missing literature, we will update the manuscript to describe mentioned studies. In the below table, we compare our method to four studies (and also in Appendix). We would like to note that our method does not use any information other than the in-distribution dataset in the training phase (e.g., class labels or other datasets).
>
> | C10 -> | R-Flow|MOOD| 1-Dim| G-ODIN| 0.0| 0.5| 1.0|
> |:--:|:--:|:--:|:--:|:--:|:--:|:--:|:--:|
> | SVHN 	| 98.2   | 96.4  | - | 98.8 |86.1|99.9|99.9|
> | TinyImgNet|  	99.6 | 		- 	| 	- 	| 	- 	| 	65.4 | 99.9 | 99.9|
> | LSUN(c)|  	- | 		99.2 | 		99.4 | 		98.3 | 		68.3|  90.9 | 98.6|
> | LSUN(r)|  	99.6 | 		93.2 | 		99.3 	| 	99.4	| 	97.0|  99.4 | 99.6|
> | ImgNet(c)| 	 - 	| 	-	| 	98.1 | 		98.7 	| 	85.2|  98.1|  98.5|
> | ImgNet(r)| 	 - | 		- 	| 	98.5 	| 	99.1 | 		85.0|  98.1|  98.9|
>
> | C100 -> | R-Flow|MOOD| 1-Dim| G-ODIN| 0.0| 0.5| 1.0|
> |:--:|:--:|:--:|:--:|:--:|:--:|:--:|:--:|
> |SVHN 	|	95.1 |		85.8 |		- 	|	95.9 |		80.9| 99.9| 100|
> |TinyImgNet |	98.1 |		- |		- 	|	- |		91.7| 100 |100|
> |LSUN(c) 	|- 	|	96.8 |		93.8 |		95.3 |		65.6 |89.9| 92.2|
> |LSUN(r) |	98.9 |		77.6 |		95.7 	|	98.7 |		97.0| 99.8 |99.6|
> |ImgNet(c)	|- 	|	- |		88.6 |		97.6 |		94.1 |100 |100|
> |ImgNet(r) |	- 	|	- |		93.7 |		98.6 |		94.7| 97.7 |94.2|

---

### Official Review · Reviewer_sj54 · 2021-11-02

**Correctness:** 3
**Technical Novelty And Significance:** 2
**Empirical Novelty And Significance:** 2
**Recommendation:** 5
**Confidence:** 4

**Main Review:**

The submission makes the case that practical applications can sometimes call purely for a detection of textural distributional shift, and at other times purely for an object-identity shift. While this certainly makes some sense, I think this perspective is a bit too narrow: all of practical OOD detection in computer vision would not fit into one of these two buckets. For example, one might be interested in the composition of scenes, where the presence of two objects that don’t usually co-occur is considered OOD. Or one might be interested in objects poses, where a car that is overturned on the road is considered OOD. This point is made in [1], where the meaning of the word “semantic” is associated with the context, thus placing every practical OOD/anomaly detection problem into first specifying context and then developing a model that captures the notion of what is semantic within that context (which would include texture, if that was the feature of interest). I believe this is a broader and more encompassing view of the point the submission is attempting to make. However, [1] did not explore beyond the context of object-identity, and this is worth exploring further.


In Eq. (1), I think one cannot drop the $p(x)$ term from the denominator, since we are modelling a (conditional) distribution on the variable $x$. $p(t, s)$, $p(t)$, and $p(s)$ may be dropped if one is not concerned with normalization, but I think $p(x)$ ought to remain since it is not uniform. It is also not completely clear to me that texture and semantics should be treated as independent, as in the MNIST-KMNIST experiment (regardless of their conditional independence given x), because a change in semantics ought to also entail a change in textures most often (since a novel object will most likely have a different texture associated with it). But I partially agree that sometimes we might want to ignore stylistic changes if the object stays the same. An earlier attempt at evaluating non-semantic and semantic shift exists [2].


From Eq. (9) it appears that \lambda = 0.0 corresponds to “semantic”-shift, but the experiments seem to suggest the reverse. Could you clarify?


I’m not sure if it is notationally accurate to say that a realNVP model on top of PCA-reduced features of an image can be referenced as the conditional distribution $p(x | f(x))$. It might be more correct to say $p(f(x) | x)$ is what’s being learned.


Method-wise, the novel contribution is really the bit about building density models on top of PCA-reduced features from existing methods for OOD detection [3,4]. The experiments seem to suggest this can lead to fairly good performance. An ablation study is required for supporting the proposed choice of initialization for multi-SVDD.


[1] Detecting semantic anomalies, Ahmed et al.

[2] Generalized odin, Hsu et al.

[3] Deep multi-sphere SVDD, Ghafoori et al.

[4] Amplitude-phase recombination: rethinking robustness, Chen et al.

[5] Transfer-based semantic anomaly detection, Deecke et al.


**Summary Of The Paper:**

The submission proposes to evaluate OOD detection problems with regards to two aspects — detect distributional shift in texture, or that in object-identity. A Fourier transform is used to identify changes in texture, and a modification of SVDD is used to identify changes in object identity, by building density models on top of a PCA-reduction of the extracted features in both cases. Experiments are conducted which showcase that the two components can detect textural vs. object-identity shift while not mis-identifying one type of shift for the other.

**Summary Of The Review:**

The submission is commendable in joining a set of recent papers [1,2,5] in pushing for properly benchmarking and defining OOD detection. The novel perspective in this particular submission seems to be the suggestion to disentangle texture from object-identity in a way that detecting one does not imply detection of the other, which is an interesting and thought-provoking view, albeit somewhat narrow. The idea of using density models on PCA-reduced features is demonstrated to work well. The submission scores low on clarity of notation.

---

> ### Author Response · Authors · 2021-11-17
> **Response to reviewer sj54**
>
> __R2-Q1) Discussion with [1] and [2].__
>
> R2-A1) We appreciate the insightful discussion about our core idea and letting us know the missing reference such as [1, 2]. We are glad to see other studies which refine the OOD task itself by dissecting and deeply analyzing the true nature of the OOD. We add it to the Related works & Introduction section.
>
> We can summarize the idea of [1] as concentrating on the semantic distribution shift to detect OOD while providing robustness against the non-semantics factor (texture, in our case). Similarly, by our OOD definition, the semantics module of our framework is robust on the non-semantics factor as well.
> In addition, our work also considers a new perspective of the OOD, which is the case where non-semantics (i.e. texture) is the critical feature to identify as the OOD, such as finding a deviated product by a crack in a manufacturing facility.
>
> As the reviewer mentioned, the diverse stylistic change scenario is interesting to discuss. In this scenario, a user may want to ignore the stylistic change, but the other possibly wants to see only the style and doesn’t matter the semantic change. For the former one, [1, 2] and our method with semantics mode are the appropriate solutions to tackle. However for the latter one, to the best of our knowledge, our novel definition of the OOD only concerns this scenario and the texture mode of our method is designed to handle such cases.
>
>
> __R2-Q2) Eq 1 (and independence of both variables).__
>
> R2-A2) Thank you for your valuable comments. As other reviewers also pointed out, we agree that the independence assumption is too strong and possibly unrealistic. Thus, we remove this and relax the assumption (as shown in Section 3.1 in our revised version). For more details, please see R1-A1.
>
> __R2-Q3) Eq. 9 ($\lambda$=0.0 corresponds to “semantic”-shift?)__
>
> R2-A3) It should be the “texture”-shift. We apologize for the confusion. We revise this in our manuscript.
>
> __R2-Q4) It is better to change the RealNVP notation from $p(x | f(x))$ to $p(f(x) | x)$ ?__
>
> R2-A4) Thanks for pointing it out. We agree with your feedback since the extracted features $f(x)$ are calculated in the flow-based model which infers the likelihood of the factors. We will revise this notation in our manuscript.
>
> __R2-Q5) Ablation study for the initialization for multi-SVDD.__
>
> R2-A5) The performance comparison of the multi-SVDD with and without our initialization strategy is as below (and we have also added this in Appendix).
>
> | ID -> OOD |	C10 -> C100	 |  C10 -> SVHN  |	C100 -> C10 |	C100 -> SVHN | SVHN -> C10 | SVHN -> C100   |
> |:--:|:--:|:--:|:--:|:--:|:--:|:--:|
> multi-SVDD |  53.7     |       99.7   |   67.5     |  84.2    | 87.5  |      84.2
> \+ Angular Init |  93.5     |       99.9   |   84.2    |   100     | 100     |    99.9
>
>
> As shown in the table, multi-SVDD with our initialization outperforms the conventional initialization-based method in the semantics-oriented scenario.
>
> __Reference__
>
> [1] Detecting semantic anomalies, Ahmed et al. \
> [2] Generalized odin, Hsu et al.

---

> > ### Comment · Reviewer_sj54 · 2021-11-24
> > **Post-rebuttal**
> >
> > Thanks for the update.
> >
> > There might be some misunderstanding with [1]: I believe the point made in [1] is that the word “semantic” need not refer to object-identity all the time, it could also refer to texture, or other things, depending on context. It is only in an object classifier that the object-identity is considered meaningful, i.e. “semantic”. So texture is not non-semantic if the task is “finding a deviated product by a crack in a manufacturing facility”. The recommendation in [1] is precisely to build a texture-based model if the task is to detect textural abnormalities (as the submission does), or an object recognition model if the task is to detect novel objects, or a scene-composition model if the task is to detect compositional anomalies.
> >
> > Thanks for the ablation study; this seems to suggest that the proposed angular initialization is indeed useful.
> >
> > Having read the reviews/rebuttal, I’m inclined to stand by my rating at this time. While I do not think the semantics/texture discussion contributes much to what is already mentioned in the literature (I find it to be narrower than in Ahmed&Courville), I think the paper might have some significant technical contributions wrt the method that is implemented and the evaluation methodology. However, at the moment I think the presentation is too messy (with imprecise attempts to use math-y derivations) and would be much more improved with a fresh, cleaner, to-the-point presentation of the technical contributions.

---

### Official Review · Reviewer_q2kp · 2021-11-07

**Correctness:** 3
**Technical Novelty And Significance:** 2
**Empirical Novelty And Significance:** 2
**Recommendation:** 3
**Confidence:** 4

**Main Review:**

My biggest concerns about this paper is the correctness of assumptions, derivations and fairness of comparison.

1. in section 3.1 'we assume that both variables have conditional independence given x'. Is it a valid assumption?
2. Equation 1 is wrong. It should be p(x|t)p(x|s)p(t)p(s) / p(x)p(t, s). The priors were ignored. And I still doubt the likelihood can be decomposed wrt. texture and semantics. For example, the texture can determine whether it is a horse or zebra, grassland or sand.
3. in section 3.2, 'p_theta(x|t) should focus on extracting texture information alone, while p_theta(x|s) solely on the semantic counterpart', the model design does not guarantee this.
4. Figure 4 shows corrupted C10 has more frequency components at 10-20, which is not surprising. I am surprised that C10 and C100 have the same spectrum. Have you tried corrupting this C10 image by alpha-blending C10 and C100 together? Or cut C10 into pieces and rearrange the spatial orders.
5. In table 1, what is the comparison of in-distribution detection of the proposed methods against the listed methods?
6. In table2&3, looks like lambda=0 performs worse than method in comparison because lambda=0 is texture mode and it is testing texture discrepancy. In table 4, when testing semantics discrepancy, lambda=1 works worse because it is semantics mode. It is unfair to compare their best models with other methods, because the methods with best lambda include the prior knowledge whether OOD is wrt. texture or semantics. In table 2&3 when you choose lambda=1, you are explicitly telling the model that OOD is wrt. texture. Other methods do not have such kind of knowledge.

**Summary Of The Paper:**

This paper decomposed out-of-distribution into texture and semantics and proposed a model that extracts the texture and semantic information separately and then combines them via normalizing flow-based method.

**Summary Of The Review:**

I have some questions that could influence my understanding of the basic blocks. I cannot make a clear decision until seeing the answers.

==========================

Based on other reviewers' comments and authors' responses, I would like to lower the score to reject this paper. The false claims and equations give me an impression the authors do not understand what they are doing. I would recommend another round of revision and rebuttal for authors to build a solid story instead of rushing this work into publication.

---

> ### Author Response · Authors · 2021-11-17
> **Response to reviewer q2kp**
>
> __R1-Q1) Assumption of conditional independence between semantics and texture variables.__
>
> R1-A1) We appreciate your thoughtful and valuable feedback on our problem statement. We agree that the previous assumption of conditional independence of semantics and texture was too strong. Instead, we revise the formulation as the descriptive score of each OOD module, i.e. lambda*logP(S(x)|x) +(1-lambda)*logP(T(x)|x). We would like to emphasize that although our assumption is relaxed, the core spirit that decomposing as semantics and textures is unchanged.
>
>
> __R1-Q2) 1) Eq 1 is wrong. 2) Justification of decomposing as semantics and texture.__
>
> R1-A2) 1) As we answered in R1-A1, we remove the Eq1. 2) In the real-world OOD scenarios, we can acquire such prior knowledge very often and easily. One case is the machine vision task in the manufacturing factory. In this circumstance, it is obvious that the OOD model should detect the cracked or fault products solely using texture (i.e. crack). On the other hand, there is a scenario that the OOD method should be robust to image corruption such as the self-driving car system.
>
> Of course, as R3-Q1 also mentioned, the texture could be some part of semantics for some datasets, the horse-zebra case that you exampled also be the case. Although both factors cannot be decoupled completely in this case, we focused on decomposing the semantics and texture to give controllability and flexibility to a user by providing the decoupled two steering wheels. For more detailed explanation, please see R3-A1. We discussed on this decomposing necessity at the 5th paragraph of the introduction.
>
> __R1-Q3) How do we guarantee that p_theta(T(x)|x) and p_theta(S(x)|x) extracts in charged factor only?__
>
> R1-A3) We admit the reviewer's concern that the semantic module does not guarantee the focus only on semantics. To solve this issue, we try to treat the semantics information as the label of the in-distribution dataset. Like [2, 3], ordinary OOD cases consider the semantic shift as the label change. So we suggest the angular distance-based initialized method can distinguish the OOD semantics. On the left side of figure 3, existing initialization methods can handle the One-class classification samples, but OOD samples are not. However, our OOD problem objective is detecting OOD semantics, not other classes in the in-distribution dataset. We believe that this cohesive trained in-distribution assumption can focus on the label information and is insensitive to texture.
>
> Empirically, we plot the embedding results of our S(x) in Figure 6 from Appendix. These figures support our assumption that S(x) separate only semantic differences.
>
> Note that the T(x) module extracts the information on frequency domain only, and the likelihood  of texture information  P(T(x)|x)  is obtained by the flow-based model. Thus, the P(T(x)|x) module can concentrate on the texture feature alone.
>
>
>
> __R1-Q4) Other corruptions on C10 (Figure 4).__
>
> R1-A4) Thank you for the interesting suggestion! In Appendix, we report the frequency visualization plot on 1) Alpha-blending between C10 and C100, and 2) Jigsaw augmentation (Figure 8 & 9 in Appendix). For the Jigsaw augmentation, we observe slight differences in the mid-level frequency due to the hard crossing border at the center of the image, while no significant change for the alpha-blend case.
>
> __R1-Q5) Comparison to the SOTA methods on the “in-distribution” detection.__
>
> R1-A5) Again, we appreciate the insightful suggestions, we haven’t thought of benchmarking such a scenario. The result is shown below (we will also update this result in revised manuscript). We evaluate the C10 test dataset by using the pretrained model on the C10 training set.
>
>
> | Dataset | ODIN | Gram | CSI | Ours |
> |:--:|:--:|:--:|:--:|:--:|
> | C10	| 47.2   | 44.6  | 61.9 | 50.1 |
>
>
> ODIN, Maha, and our method perform 50% AUC (is able to distinguish in-distribution as in-distribution) while CSI shows 61.9%. We suspect that CSI overfits the C10 training dataset.
>
> __R1-Q6) Is this a fair comparison that provides prior knowledge to our OOD model (i.e. lambda=0 or 1)?__
>
>
> R1-A6) Again, We would like to emphasize that the prior knowledge of the given OOD detection task is realistic. Please see R1-A2 for more details about the prior use case examples.
>
> Compared to dealing with the semantic distribution shift in OOD [2. 3], our framework can handle the prior knowledge. Moreover, As the reviewer commented, there could be scenarios that prior knowledge cannot be acquired. In this case, a user could choose lambda=0.5 (no prior) and this model achieves reasonable and comparable performance to SOTA in most cases.
>
>
> [1] ImageNet-trained CNNs are biased towards texture, Geirhos et al. \
> [2] Detecting semantic anomalies, Ahmed et al. \
> [3] Generalized odin, Hsu et al.

---

> > ### Comment · Reviewer_q2kp · 2021-11-23
> > **Comments to authors**
> >
> > 1) 'Note that the T(x) module extracts the information on frequency domain only, and the likelihood of texture information P(T(x)|x) is obtained by the flow-based model. Thus, the P(T(x)|x) module can concentrate on the texture feature alone'
> > T(x) can extract any other information except semantics, not necessarily texture. For example, the high frequency components represent object boundaries, ie. the shape of the object.
> >
> > 2) From the ablation studies, lambda=0.5 does not show clear advantage. The performance is either inferior to or only slightly better than other methods in comparison.
> >
> > 3) Since the authors mess up the correspondence between lambda=0/1 and texture/semantics mode, the story the authors were telling in the ablation study does not hold.

---

### Author Response · Authors · 2021-11-17
**Summary of the revision.**

We would like to thank all four reviewers for their valuable feedback. Their fair criticism and constructive suggestions have enabled us to improve the quality of our manuscript.

We update the Appendix and revise the papers to summarize the following items.

+ Relax previous conditional independence assumption and revise this more clearly in Section 3.1.
+ Add ablation study on our novel angular initialization method in Appendix (Table 2).
+ Add experiment results on in-distribution detection tasks in Appendix (Table 3).
+ We add an additional OOD benchmark in Appendix (Table 1).
+ We visualize embedding plots of Multi-SVDD with our angular initialization in Appendix (Figure 2) to empirically show how our semantic module, S(x), is able to extract semantics effectively.

The revisions are marked with "red" in the manuscript.
We proceed to answer the points raised by each of the reviewers individually below.

Best regards, Authors

---

### Decision · Program_Chairs · 2022-01-20

**Decision:**

Reject

**Comment:**

This paper proposes a categorization of out-of-distribution examples by texture and semantics, and proposed a model that extracts the texture and semantic information separately before combining them via a normalizing flow-based method to obtain good results. While the categorization provides some interesting perspectives, most reviewers found the assumptions too strong, and there are some issues with the derivation. Reviewers have some positive feedbacks on the proposed algorithm for OOD, but also expressed concerns about the fair comparison with more recent baselines. The paper, in its current form, is not ready for the publication, but the authors are encouraged to improve the paper with reviewers' suggestions and resubmit.